# Real-data-driven real-time reconfigurable microwave reflective surface

Erda Wen [1] ✉, Xiaozhen Yang [1] & Daniel F. Sievenpiper[1]

Manipulating the electromagnetic (EM) scattering behavior from an arbitrary surface dynamically on arbitrary design goals is an ultimate ambition for many EM stealth and communication problems, yet it is nearly impossible to accomplish with conventional analysis and optimization techniques. Here we present a reconfigurable conformal metasurface prototype as well as a workflow that enables it to respond to multiple design targets on the reflection pattern with extremely low on-site computing power and time. The metasurface is driven by a sequential tandem neural network which is pre-trained using actual experimental data, avoiding any possible errors that may arise from calculation, simulation, or manufacturing tolerances. This platform empowers the surface to operate accurately in a complex environment including varying incident angle and operating frequency, or even with other scatterers present close to the surface. The proposed data-driven approach requires minimum amount of prior knowledge and human effort yet provides maximized versatility on the reflection control, stepping towards the end form of intelligent tunable EM surfaces.

It has been decades since the idea was first proposed to control EM field behaviour with metamaterials—structures with subwavelength geometrical details[1–3]. In particular, its 2-D version, i.e., metasurface, draws broad attention and is intensively investigated due to its advantage in engineering aspects—being able to be manufactured relatively easily on thin sheet materials[4]. The reported application space of metasurfaces is vast, ranging from directing surface waves in the near-field[5], beam-forming in the far-field[6], to cloaking[7] and holography[8,9], etc. Beyond the planer regime, efforts have been made to implement flexible metasurfaces in hope of bringing these intriguing wave manipulation capabilities to surfaces with arbitrary shapes. However, the mechanism of wave interaction with curved surfaces is significantly more complex than its flat counterpart[10], and as a result, research has mainly focused on optimizing for specific tasks such as wave-front control[11], radar-cross-section (RCS) reduction[12] or polarization conversion[13]. To realize a reconfigureable version is even more challenging. To start with, the severe RF loss of thin, electrically tunable metasurface generally limits its application to less loss-sensitive tasks in microwave frequency[14–16]. In addition, there is a lack of accurate and efficient algorithms to support the real-time inverse design.

Meanwhile, recent years have witnessed the emergence of exploiting neural networks (NNs) in complicated EM/photonic systems. On the one hand, as a good regressor of highly non-linear functions, NNs provide a cost-efficient solution to many analysis problems, from solving Poisson's equations[17], to handling inverse EM scattering problem[18]. On the other hand, recent research demonstrates the strong design capability of NNs, including optimizing linear phased arrays[19], or designing photonic devices and nanoparticles[20–24]. One special advantage of a pure NN-driven scheme, compared to conventional optimization methods, is that no iterative process is involved in the prediction phase, which is crucial for an on-site system in need of fast response. This feature has been exploited recently to facilitate an emerging concept—the intelligent metasurface, which refers to metasurfaces that tune themselves in an adaptive manner, with little human intervention[25], from beam-forming[26,27], to sensing purposes[28,29]. In theory, similar strategy can be used for curved surfaces, and preliminary studies on NN-driven non-planer surface have been reported for cloaking or illusion applications[30–32], yet a more universal and versatile scheme is still needed for the surface to dynamically operate under different types of tasks.

[1]Department of ECE, University of California San Diego, La Jolla, CA, USA. ✉e-mail: ewen@ucsd.edu

This Article aims to demonstrate one possible universal NN-driven scheme to realize an intelligent surface that can respond to arbitrary design goals. We start by demonstrating a practical realization of a tunable conformal coating operating at microwave frequencies, whose reflection pattern can be controlled with multi-channel bias voltages. A sequential neural network architecture is then proposed, which can take free-form design targets on the pattern and environmental factors as the input. We demonstrate that for reconfigurable EM design, it is very feasible to collect and utilize real measurement data, which turns out to be fast, accurate, and very adaptive to different environments compared to simulated or calculated dataset.

## Results

### Reconfigurable flexible metasurface

The conformal metasurface design is based on a classical tunable reflective metasurface topology in ref. 33. The surface is tiled with sub-wavelength metallic patches, with varactor diodes placed between the neighboring unit cells. By changing the reverse bias voltage across a varactor diode, the reflection spectrum of a unit can be shifted. This results in tuning the local reflection phase in a frequency range close to the resonance, and collectively all units form a reflection pattern in the far-field region.

While this continuous-phase tuning approach generally provides greater degrees of freedom than discrete-state tuning methods like binary phase states with pin-diodes (also known as phase coding)[34], it is more sensitive to dissipation loss which leads to significant decrease in reflection amplitude near the resonant frequency. In supplementary note 1 we show how it is especially problematic for designs with smaller physical volume, and why thicker substrate is preferred to maximize the radiation efficiency of the metallic patches. This is particularly unfavorable for conformal design since thick radio frequency (RF) low loss materials are not typically flexible. To address this we propose a double-layered rigid-flex stacking structure to provide the overall flexibility of the coating: unit cells are implemented using relatively thick microwave materials, which are separately attached to a single ultra-thin flexible layer that also contains circuits for bias feeding. In

this demonstration, we built 24 separate columns, with 10 patches in each column, making a 38.51 cm × 13.64 cm × 1.97 mm surface. Units on each column share the same reversed bias voltage, forming a 24 dimensional vector **V**, realizing pattern control in the azimuth plane, with intensity noted as $D(\theta)$.

Figure 1c, d depict the measured static reflection amplitude/phase response of a flat board. Note that this data is not used throughout the entire inverse design workflow but rather merely as an initial examination and verification of the surface reflection performance. The results show that the board covers a large reflection phase range within the 4.5–4.7 GHz frequency band that is necessary to achieve maximum pattern tunability.

Determining the relationship between the reflection pattern and a given bias combination $D(\theta) = f(\mathbf{V})$ can be challenging, due to the non-linear relationship between bias voltages, local reflection phases and directivity to certain directions. Additionally, the coupling and multi-reflection effect in the concave regions invalidates any theoretical models that consider the surface as simply a reflective antenna array[31] (Supplementary Note 3). Other factors that complicate the problem are, but are not limited to, the reflection phase that depends to different incident angle for each column (Supplementary Note 3), tolerance of individual lumped components, or in some cases, the presentation of scattering objects near the device. In this scenario, a pure data-driven model is especially advantageous, since it can automatically take into account of all these factors by using the real measured data. However, the extremely large search space of the input variables renders the conventional interpolation or regression methods impractical and make the neural network method the best candidate.

### Sequential tandem neural network

Consider the fact that a simple feed-forward network (FFN), or a multilayer perceptron (MLP), can theoretically approximate any given function provided large enough scale, it is tempting to believe it can be used to find the underlying pattern-voltage mapping $\mathbf{V} = f^{-1}(\mathbf{D})$. The pitfall is that the design parameters and design goals usually have a

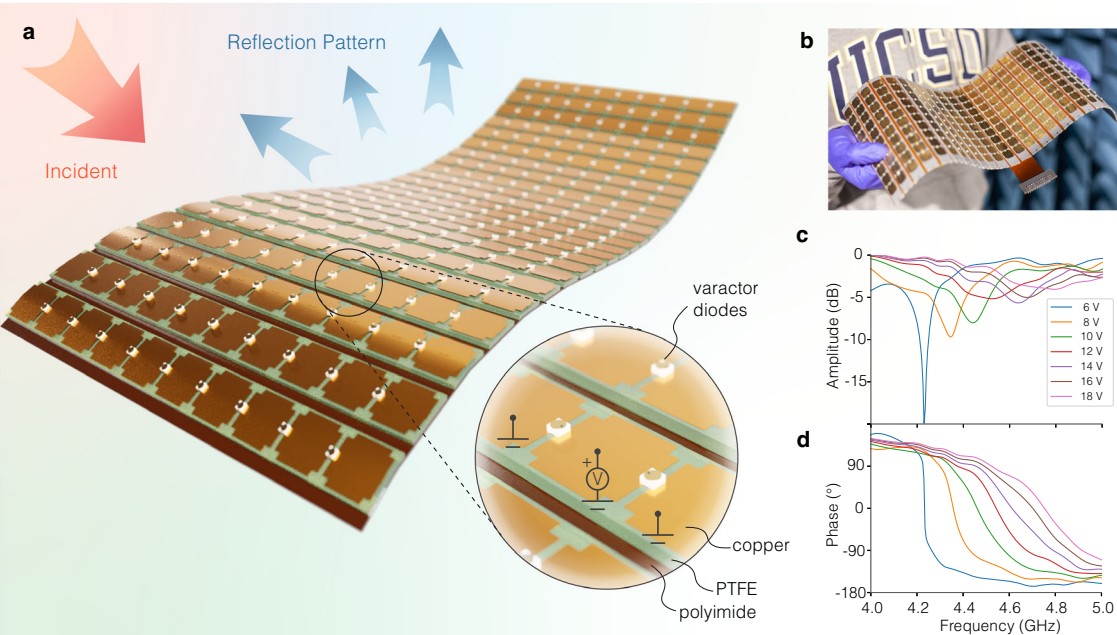

**Fig. 1 | Conformal metasurface and its static performance. a** Illustration of the conformable rigid-flex printed circuit design. The substrate of each columns is made with 1.57 mm semi-rigid reinforced PTFE material designed for microwave applications. They are then bonded to a single flexible sheet made of 0.18 mm thick polyimide, with a feeding network on the bottom side to provide the varactors with d.c. bias voltages. **b** Photo of the prototype. **c, d** Measured reflection amplitude and phase response under normal incidence of the flat surface under various bias voltages.

multiple-to-one relationship. In this case, similar or even identical reflection patterns may correspond to very different bias voltage combinations. When the network is trained with gradient descent methods, conflict gradients may arise from data with very similar input (i.e. pattern) but very different labels (i.e. bias voltages), preventing the parameters from converging. For a problem where the input contains less information than the output, as in this case, a generative type of neural network is necessary. Recent large generative models such as diffusion[35] and transformer[36] have shown extremely powerful capability in generating image and language content, even stepping towards artificial general intelligence (AGI)[37], yet for engineering problems on specific tasks, small-scale efficient models are still preferred, among which the tandem architecture has proven a very effective framework[20,21,38–40].

In tandem architectures, a predictor is first trained to solve the analysis problem, in our case, the bias-pattern mapping; then another network, the desinger, is trained to handle the synthesis procedure—determining the bias combination given a specific pattern. The designed bias can be fed into the predictor to produce an expected pattern $\tilde{D}$, and the performance can be evaluated by comparing the discrepancy between $\tilde{D}$ with the design goal $D$, which serves as the loss function for training designer network. Importantly, the second step does not involve $D - \mathbf{V}$ mapping from any dataset, thus there is expected to be no gradient conflict. Essentially, instead of fitting the reverse function $f^{-1}(\cdot)$, the network aims at seeking any function that simply optimizes the design performance.

Notice in conventional tandem networks reported by previous works, a design target with the exact form as the predictor's output is required as the input of the network, for example, a vector that contains the whole pattern with a certain resolution. This largely limits its practicality since in many cases, free-form design goals are more favorable, for example, one may want to specify several target directivity intensity in certain directions without having to constructing the entire pattern. To enable this free-form input capability, here we introduce the recurrent neural network (RNN) layer in the designer Fig. 2. RNN is typically utilized to process temporal signals such as video and speech[41]: the recurrent layer updates itself from a current state as a sequential signal is fed in, resulting in a memory effect on all

past inputs in the sequence. Here we can use a sequence of design goals as the input, which could be, for example, a sequence with a length of $l_t$, repeating $n_t$ angle-directivity pairs $(\theta_i, D_{\theta_i})$. Despite that there is no explicit temporal relationship between these design goals, we still expect the layers to memorize all those targets within the sequence. In this way, the network can respond to design goals with arbitrary dimensions as long as $n_t$ is below reasonable threshold to avoid vanishing gradient.

For the predictor, convolutional layers are employed, which is based on the physical knowledge of a linear phased array[42]: the same phase difference between neighboring units should have similar effect in the far-field no matter where they are located within the array, and therefore the parameters can be shared among all adjacent units. The convolutional layers significantly reduce the number of parameters so that the predictor requires less data and suffers less from overfitting.

To allow the surface to operate with changing incident frequency and under different incident angles, this varying environmental information can be also cascaded to the input design goal vectors in the designer, and to the input bias-voltage vectors in the predictor.

## Experiments and Results

Here we demonstrate four scenarios in which the conformal metasurface may operate, with an increasing complexity as follows: A) the simplest case of a flat surface operating under a normally incident plane wave at a single frequency; B) a curved surface, under normal incidence, at a single frequency; C) a curved surface working in a varying environment: with incident wave angles ranging from − 30° to 30°, and a frequency band from 4.5 GHz to 4.7 GHz, and D) a curved surface under varying incident angle and frequency, with a plastic scattering object present in front of the surface, disturbing the reflective pattern. The pattern data are gathered using a setup in an anechoic chamber shown in Fig. 3. The patterns are collected in 5-degree-resolution from 0° to 180°, forming a 37-dimensional vector **D**.

For cases with constant environment (case A and B), 20,000 samples are collected with random combinations of bias voltages ranging from 0 V to 18 V on each channel. For case C and D, 4000 random bias samples are collected for each incident angle, with 13 incident angles; 5 random frequencies within the band of interest are

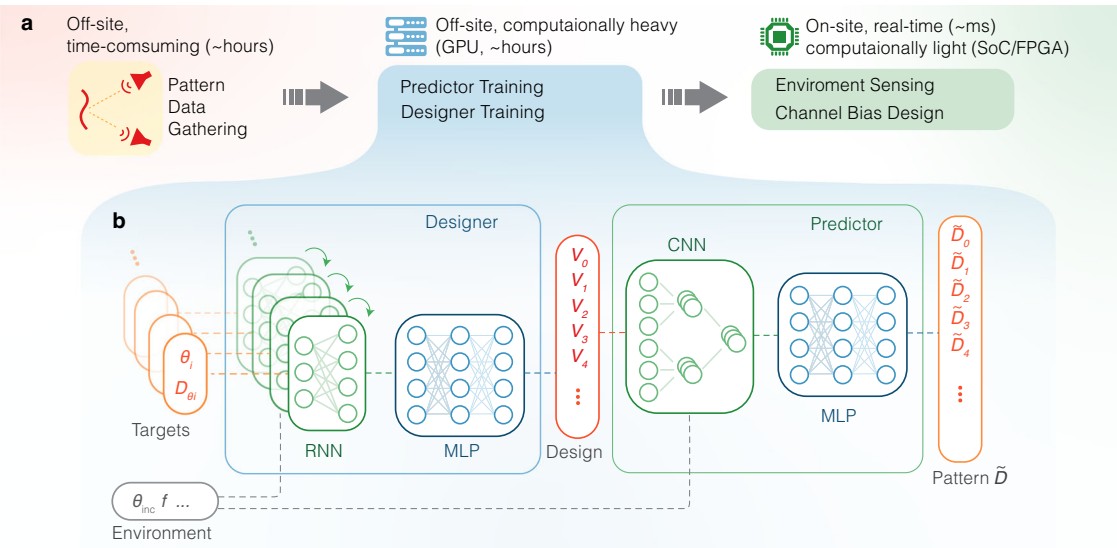

**Fig. 2 | Real-data-driven real-time inverse design workflow. (a)** The time-consuming and computationally-heavy part are done off-site in the first two steps - data gathering and network training. The pre-trained network can be then deployed to on-site controllers with very limited computing resources to realize fast-response inverse design. **(b)** Proposed sequential tandem network architecture.

The predictor is first trained with measured pattern data. Then the parameters of the predictor are fixed and random design target sequences are used to train the designer. Detailed layer dimensions and connections are shown in Fig. S9 in Supplementary Note 5.

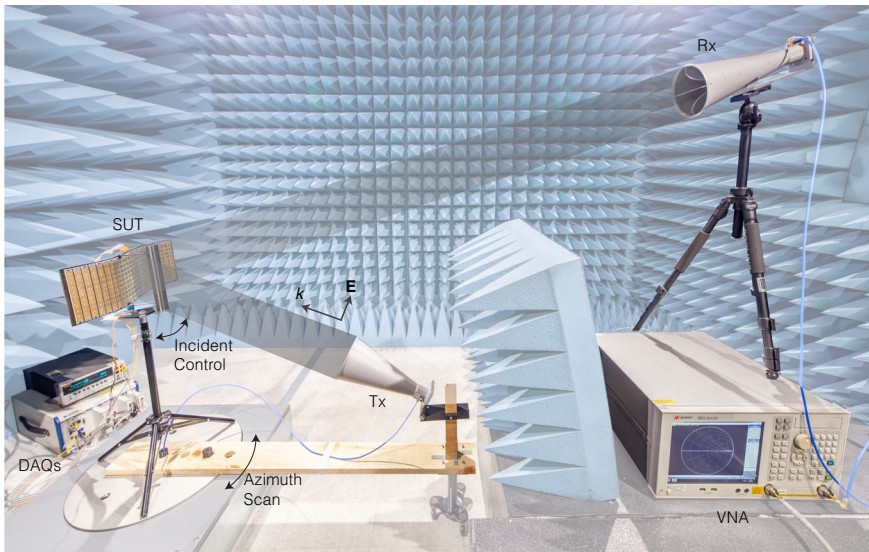

**Fig. 3 | Experiment setup for pattern data gathering.** The data collection is done in an microwave anechoic chamber but can be also done in environment most resembles specific real-world condition as needed. A vertically polarized beam is excited with a horn antenna Tx, and its specular reflection from the surface under test (SUT) is received by another horn antenna Rx. The intensity and phase are recorded by a vector network analyzer (VNA). Both SUT and Tx antenna are attached to a servo motor to realize azimuth pattern scanning. Another servo motor is used to rotate the SUT to simulate incident angle changes. Revered bias-voltages of 24 channels are generated and applied to the board with data acquisition (DAQ) cards. The whole data gathering process is automated controlled by a single National Instruments controller.

sampled for each incident/bias setup, making a total 260,000 samples. It is worth noting here that the number is absolutely not comparable to dataset needed for, say, building a table lookup table: a most coarse grid search with a resolution of 0.2V (very reasonable from the reflection curve) from 10V to 18V makes $40^{24} = 2.8 \times 10^{38}$ samples. As the response time of the varactor diodes is relatively fast, on the order of nanoseconds, the data collecting speed is mostly limited by the response time of the control/measurement instrument being used, generally on the order of milliseconds. To obtain a stable result for our setup, 20 ms wait time is used in between samples so the total data collection time is on the order of several hours, details listed in Table 1, which is much faster than any full-wave simulation methods can achieve.

The data is split 80/20 as the training/test set to train and evaluate the predictor. Fig. S11 in Supplementary Note 5 shows the performance of the trained predictor network. The prediction matches extremely well with measured data in test set for all four cases, with an error almost close to the noise level in the chamber, and considering the signal to noise ratio (SNR) is estimated to be above 30 dB (Supplementary Note 4), the trained predictor provides great accuracy for the following-up designer training.

Training the designer network is a non-supervised learning process, since the label for any design targets is itself: the loss function is defined by the masked means square error (MSE) on target directions $L(\mathbf{T},\tilde{\mathbf{D}}) = \frac{1}{n_t}\sum_{i=1}^{n_t}(D_{\theta_i} - \tilde{D}(\theta_i))^2$. In this demonstration, we randomly generate sequences with up to five targets. In practice, we found the sequence length $l_t$ should be three or four times of target number $N_t$ to yield a converged output, thus for up to 5 targets, we choose $l_t = 20$. Considering the energy distributing effect for multiple targets, the directivity ranges for different target numbers $n_t$ is $[0, D_{max}/\sqrt{n_t}]$, where $D_{max} = 8.85$ is the maximum directivity that can be ideally achieved with the surface aperture. For cases A and B, 100,000 samples are generated, and for case C and D, 260,000 samples. The data is again split 80/20 for training/test set, and the performance is shown in Fig. 4. The network performs very well for fewer numbers of targets and still decently well for 3 or more targets, obtaining an average RMSE below 1 for most cases.

It is worth noticing that this performance is evaluated on random targets that is not necessarily physically feasible, such as the existence of a peak and a null in proximity, or strong beams at the end-fire direction.

Compared with the training process, the prediction of the network consumes minimal computing resources. Therefore, this trained network can be deployed on modest micro-controllers with very limited computing power. In Table 1 we demonstrate the speed of inverse design (with designer) and the speed of evaluation (with predictor) in addition to inverse design on a controlling system using a cheap commercially available SoC controller Raspberry Pi. The speed depends on various factors such as the machine learning platform, batch sizes of input, etc, but generally the responding time for both designing (with designer) and evaluation (with predictor) are on the order of milliseconds per sequence, which can be considered as real-time.

## Discussion

Being able to specify target directivities in multiple directions makes the surface suitable for numerous applications. One simple example is to reduce the back-scatter of a surface for manipulating RCS in a dynamic environment—by specifying a null at the direction of

## Table 1 | Time consumption of tasks in the workflow

|  |  | Case A | Case B | Case C | Case D |
|---|---|---|---|---|---|
| Data gathering |  | 5h26min | 5h27min | 14h37min | 14h32min |
| Network training[1] | Predictor | 5min | 18min | 27min | 26min |
|  | Designer | 46min | 1h1min | 2h5min | 3h44min |
| Network prediction[2] | Inv. design | 5.3ms | 5.5ms | 7.7ms | 7.7ms |
|  | Inv. design + evaluation | 5.9ms | 6.0ms | 10.3ms | 9.8ms |

[1]Server hardware specs are described in Methods, network training process in Supplementary Note 5.
[2]On a lite micro-controller Raspberry Pi 4, specs given in Methods.

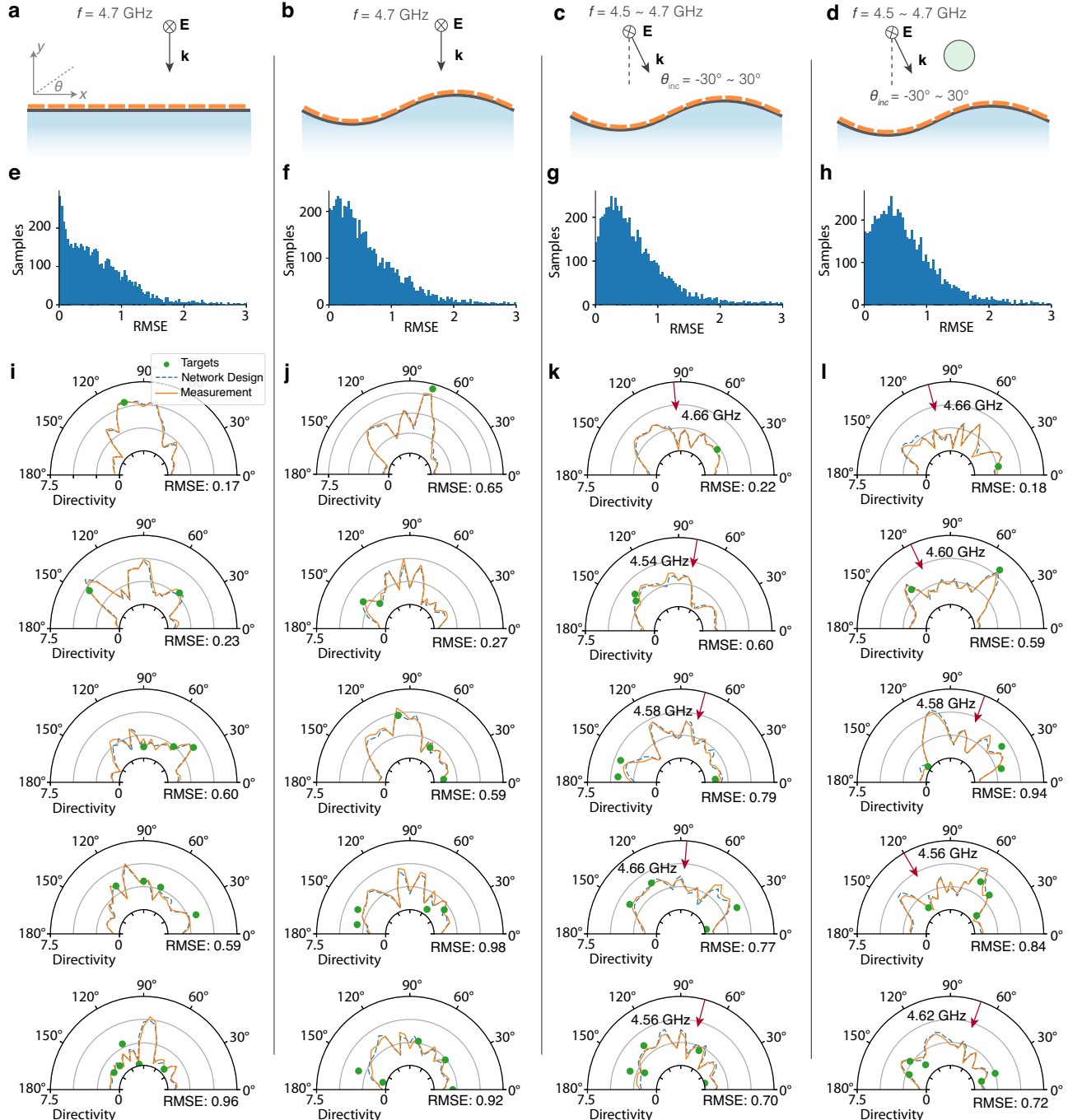

**Fig. 4 | Performance of the conformal surfaces inverse design with increasing complexity. a**, **b** Flat and curved surface, under normal incidence, at constant frequency 4.7 GHz. **c** Curved surface, under varying incident angle and frequency and **d** with scattering object presented. **e**–**h** The root mean squre error (RMSE) distribution on designs in the test set. **i**–**l** Visual examples of pattern design with 1 to 5 design targets, with typical error value in its category, excerpted from Fig. S13–S16 where more random-selected visual examples and error distribution in different categories are given. Red arrows in case C and case D indicate the incident direction.

the incoming wave, the surface can be constantly optimized to reduce the mono-static RCS to a single station, as shown in Fig. 5a. A more intriguing task is to instruct the surface to cancel out the scattering from an object in front of the surface, keeping it from being detected in certain directions, as in Fig. 5b. It can also be utilized for intelligent communication applications, performing tasks ranging from creating a single pencil beam, as in Fig. 5c, to arranging multiple peaks and nulls, completely redistributing the incoming energy as in Fig. 5d. In Fig. S17, we also give more examples for beams/nulls steering, the most common design targets in real world scenarios.

The design presented in this paper works over relatively narrow bandwidth but this is not limited by the model itself: design para-meters like substrate thickness can be increased to increase the bandwidth. The efficiency can be improved as the semiconductor technology for varactors develops. The double stack methodology can be also used for any other units design such as in[30], which is more suitable to be accommodated to a 2-D surface in the future due to the component placement (Supplementary Note 6). Modifications can also be made to the network input to achieve even higher versatility, for example by expending the target pairs with operators to form

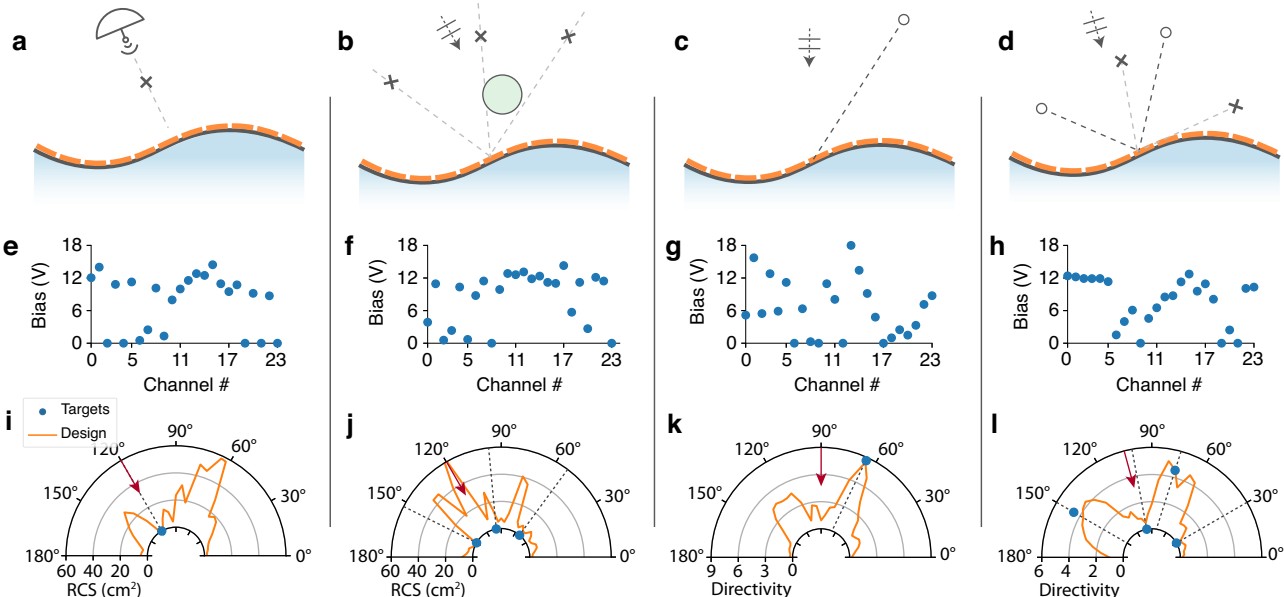

**Fig. 5 | Exemplary applications of the free-form design capability. a** Creating minimal back-scatter for a 30° incident. **b** Creating nulls in 50°, 100°, and 150° directions to stealth a scatterer in front of the surface. **c** Creating a pencil beam at 65° direction under a normal incident. **d** Creating beams at 75° and 150° direction and nulls at 30° and 100°, under 15° incident. The operating frequency is 4.6 GHz for these four examples. **e–h** Bias voltages on channel 0–23, designed by the designer. **i–l** The corresponding patterns.

tokens like $(\theta_i, D_i, \leq)$ representing design goal $D(\theta_i) \leq D_i$. Other types of tasks, such as polarization conversion or holographic pattern, can also be achieved with proper types of surface coating design, by including polarization or near-field data in the input and output of the network. Other possible future study may include enabling the coating to work on moving parts with constantly varying shapes, by parameterize the geometry and include that information into the network.

Considering the versatility of the proposed workflow, we believe this work paves the way for the next generation of tunable EM devices working under extremely complex and dynamic environments. In addition, the proposed sequential tandem architecture can potentially be accommodated to any real-time inverse design problem in different science/engineering fields.

## Methods

### Rigid-flex PCB manufacturing
The rigid layer is made with Rogers RT/Duroid 5880 material and the flexible layer is made with polyimide substrate,The unit periodicity is 13 mm along the column and 16.05 mm across the column with 3.05 mm separation in between columns (Supplementary Note 1). GaAs Hyperabrupt high-quality-factor varactor diode Macom MA46H070-1056 are used across the units. Each channel is protected by a 10 $k\Omega$ series resistor RNCF0603TKY10K0 by Stackpole Electronics Inc.

### Measurement setup
The static reflection spectrum of a flat metasurface is measured with a single horn RCDLPHA2G18B by RF-Lambda. $S_{11}$ is recorded with an Agilent E5071C VNA for 3 cases: (1) bare horn, (2) metatasurface in front of the horn, and (3) an aluminium plate of the same size in front the horn. The effective reflection amplitude and phase are then calculated (Supplementary Note 2).

The example curve in case B,C and D is a spline with 4 anchor points. The plastic scatter in case D is a 3-D printed PLA cylinder (Supplementary Note 3).

The pattern collection test bench is controlled by a single controller PXIe-8135 by National Instruments(NI) running python 3.7. For the bias voltage supply, three 8-channel 16-bit DAQ cards NI PXI-6733 and a d.c. source Keithley 2410 are used. The azimuth scanning is

realized with an ETS Lindgren 2005 motor and the incident control is facilitated by a ZOSKAY DS3235SG servo motor, driven by an Adafruit FT232H breakout and a Adafruit 12-bit PMW driver. Directivity and RCS are calculated with respect to the measured reflection from an aluminum plate of the same size as the board. Blank case calibration and time-gating are used in post-processing to reduce noise and undesired reflection from the experiment setup (Supplementary Note 4).

### Neural network and training process
The neural network is implemented using TensorFlow 2.6.0 under Python 3.9.5, and is trained on the UCSD Data Science/Machine Learning Platform (DSMLP) using four Intel XeonGold 6130 CPU @2.1 GHz core and one NVIDIA GeForce GTX 1080 Ti GPU.

The designer consists of 3 recurrent layers and 3 flat fully connected layers. The predictor consists of 3 convolutional layers and 3 flat fully connected layers. The parameters are trained with ADAM[43] optimizer. L2 regularizaion is used for the predictor to reduce the overfitting[44]. See Supplementary Note 5 for the detailed training process. The SoC computer for network prediction speed evaluation is Raspeberry Pi 4 with Quad-core Cortex-A72 @1.8GHz and 8GB RAM, running Python 3.7.3 and TensorFlow lite 2.3.0. The time is the average on a dataset with 6000 samples for case A,B or 6500 smaples for case C,D, processing with a batch size of 500.

For visual examples in Fig. 4 and Fig. S13–S16, we use random seeds 1,2,3,4 when sampling for case A,B,C and D, respectively. to ensure the generality and reproductivity.

## Data availability
Raw pattern data and processing code to generate training samples for the neural network has been deposited to https://doi.org/10.6084/m9.figshare.22908602. Source Data for Fig. 1c, d, Fig. 4e–h, Fig. S13–S16b, from which Fig. 4i–l are excerpted, and Fig. 5e–l, are provided with this paper.

## Code availability
The example Python code (in Jupyternote format) for building and training the network is provided on https://github.com/ErdaWen/Real_data_driven_Metasurface.

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

## Acknowledgements

This work is supported by Air Force Office of Scientific Research (AFOSR) under grant FA9550-21-1-0167 and National Science Foundation (NSF) under grant 2148318.

## Author contributions

E.W. and D.F.S. conceived this work, E.W. fabricated the samples and established the NN model, E.W. and X.Y. performed the experiment, and D.F.S. directed and supervised the research. E.W., X.Y., and D.F.S drafted the original manuscript.

## Competing interests

The authors declare no competing interests.
