## [Peer Review File · Nature Communications]

Real-data-driven Real-time Reconfigurable Microwave Reflective SurfaceREVIEWER COMMENTS

Reviewer #1 (Remarks to the Author):

In this manuscript, the authors demonstrated an interesting way to design adaptive metasurfaces operating in microwave frequencies with a real-data-driven tandem artificial neural network. A 24-column 1D reconfigurable metasurface is fabricated for real-data acquisition. A generative-type neural network is firstly trained using some of the collected experimental data (80%), and then the performance of the trained neural network is tested using the rest of the data. The number of design targets can be as much as 5, and overall the performance of the device is very similar to the network design, confirming the great validity of the proposed method.

As compared with the previous result, this manuscript focuses on a regime (e.g, non-flat surface with a scatter near the surface) where the rational design method is hard to guarantee a good performance. From this point of view, this idea contains enough novelty and should be of great interest to researchers working in the field of optical communications and photonics. However, I would like to ask the authors to address the following questions before making the final decision:

1) I wonder if the authors can provide a numerical study to reveal how "bad" the conventional metasurface design method is in the situations reported in the text (i.e., non-flat surface, or perturbed scatter existed). If we just assume that there is no crosstalk between meta-atoms, we can map the target function with the meta-atom configuration with a simple Fourier transform. Of course, such a configuration may not work very well, but I hope to see how "bad" it is if we use the neural-network-predicted design as the bench marker.

2) The authors focus on free-form designs in the manuscript, but I would say the well-defined target function could be more important in phased-array technologies. I am sorry to say that, but to me, most of the designed far-field patterns seem like essentially Lambertian scattering with "random" noise, and I can hardly imagine and believe such a specific far-field pattern is the optimized solution for certain applications. I strongly recommend the authors use single-beam, dual-beam, and tri-beam steering as the target function to illustrate the power of the reported design principle. These are very basic, important, and easy-to-evaluate applications for phased-array technologies.

3) The authors demonstrated a 24-column 1D adaptive metasurface in the manuscript, which is good enough for a proof of concept. However, if we want to use this concept for practical, real applications, the dimension of the vector will become much larger (>1000). I guess that the computational cost is not growing linearly with the number of dimensions, and I wonder if the author could do some simple calculations and comment on the scalability of the proposed method.

4) The authors claimed the advantage a few times that they use measurement results instead of simulation results for data acquisition. My understanding is that people use simulation only when it is more convenient than doing real experiments. For example, much EM research is focused on passive device design, and it is impossible to fabricate millions of samples and test their performance one by one. I believe people make different choices in different situations, and the logic of some statements about this point in the manuscript could be improved. Perhaps the best way is to simply say "As we work with a reconfigurable design, actual experimental data is more convenient to be used for data acquisition, avoiding any possible errors that may arise from calculation, simulation or manufacturing tolerances."

Reviewer #2 (Remarks to the Author):

This paper presented a deep-learning-enabled reconfigurable conformal metasurface. A flexible reconfigurable metasurface is presented and tested. The recurrent neural network is applied to achieve a neural-network predictor with customize input-size. However, considering the high quality of Nature Communications, I do not find enough innovation in this paper.

The authors declared that the reconfigurable flexible metasurfaces for curved surfaces are very challenging. However, some similar works have been achieved many years ago [1-4]. The metasurface designed in this paper does not show any advantages in terms of amplitude loss and bandwidth. More recently, the flexible RF circuits with more compact size and functions have been proposed [5,6]. Thus, I do not find innovation of the metasurface designs in this paper.

Another advantage declared by the authors is using the measured data for neural network training (called as 'real-data-driven'). As the numeric simulations require matrix calculations, the direct measurement is surely faster than the simulations. However, the idea of using real-data is not new and has already been used for various reconfigurable metasurface imaging and radar detections [7, 8]. The authors did not demonstrate more breakthrough methods and theories in this paper..

Two advices:

1. The symbolic representation in this paper is confusing. In page 5, "a sequence with a length of l_t , repeating n_t angle-directivity pairs (θ_i, D_i) ", the definition of subscript 't' and 'i' should be explained. In Fig.2b, the input of the designer is $(\theta_i, D_{\theta i})$, however, in Extended Data Fig. 1, the input of the designer is (θ_i, D_i) , which is not uniform. Meanwhile, Fig. 2b displays an autoencoder structure made of designer and predictor, which means the output of the predictor (D_i) should correspond to the input of designer $(\theta_i, D_{\theta i})$. The authors should explain the correspondence between D_i and $D_{\theta i}$, better shown in the figure.

2. In page 8, the authors mentioned that "Therefore, this trained network can be deployed on modest micro-controllers with very limited computing power." However, the efficiency of the designer network hasn't really been verified in a micro-controllers nor embedded in a system-level application. As reference works, a deep-learning-enabled self-adaptive metasurface clock deployed on micro-controllers is displayed in [R1], and an intelligent metasurface system for automatic tracking of moving targets and wireless communications deployed on FPGA is displayed in [R2]. The authors should add [R1] and [R2] in the References and discuss the advantages of this paper compared with them.

[R1] QIAN C, ZHENG B, SHEN Y C, et al. Deep-learning-enabled self-adaptive microwave cloak without human intervention. *Nature Photonics*, 2020, 14(6): 383-+.

[R2] LI W, MA Q, LIU C, et al. Intelligent metasurface system for automatic tracking of moving targets and wireless communications based on computer vision. *Nature communications*, 2023, 14(1): 989-.

References:

[1] Park, E., Li, W., Jung, H., Lee, M., Park, J.-H., Shamim, A., Lim, S., *Adv. Mater. Technol.* 2023, 8, 2201451.

[2] X. -Y. Luo et al., "Active Cylindrical Metasurface With Spatial Reconfigurability for Tunable Backward Scattering Reduction," in *IEEE Transactions on Antennas and Propagation*, vol. 69, no. 6, pp. 3332-3340, June 2021, doi: 10.1109/TAP.2020.3037728.

[3] X. D. Liang and A. J. Crosby, "Uniaxial stretching mechanics of cellular flexible metamaterials," *Extreme Mechanics Letters*, vol. 35, Feb 2020, Art no. 100637.

[4] S. Liu, H. Xu, H. Zhang, and T. Cui, "Tunable ultrathin mantle cloak via varactor-diode-loaded metasurface," *Opt. Express* 22, 13403-13417 (2014).

[5] G. Li et al., "Three-dimensional flexible electronics using solidified liquid metal with regulated plasticity," *Nature Electronics*, vol. 6, no. 2, pp. 154-163, 2023/02/01 2023.

[6] Y. Bai et al., "A dynamically reprogrammable surface with self-evolving shape morphing,"

Nature, vol. 609, no. 7928, pp. 701-708, 2022/09/01 2022

[7] L. Li et al., "Machine-learning reprogrammable metasurface imager," Nature Communications, vol. 10, no. 1, p. 1082, Mar 6 2019, Art no. 1082, doi: 10.1038/s41467-019-09103-2.

[8] S. Wei et al., "3DRIED: A High-Resolution 3-D Millimeter-Wave Radar Dataset Dedicated to Imaging and Evaluation," Remote Sensing, vol. 13, no. 17, 2021, doi: 10.3390/rs13173366.

Reviewer #3 (Remarks to the Author):

Manipulating the electromagnetic (EM) scattering behavior dynamically is an ambitious objective in the fields of EM stealth and communication. However, conventional analysis and optimization techniques fall short in accomplishing this task. In this paper, a reconfigurable conformal metasurface prototype is introduced, capable of responding to various design targets in reflection patterns with minimal on-site computing power and time. The metasurface utilizes a sequential tandem neural network that is pre-trained with real experimental data, eliminating errors from calculations, simulations, or manufacturing tolerances. I conclude the innovations of this work as two parts, the realization of conformal metasurfaces and on-site machine learning. I need to make further consideration after the authors solve the following problems.

-The tunable conformal metasurfaces are demonstrated in one dimension. How to realize two-dimensional case, which is more useful.

-Tandem neural network has been studied before to imitate the non-uniqueness issue in inverse design. In this work, what's the new points in sequential tandem neural network.

-For each scene, the authors need to train the neural network case-by-case. Although the authors clarify that they use experimental data and on-site training, the case-by-case neural network is inefficient. In other words, if it is possible to build up the neural network connection between different scenarios to speed up network convergence and save the amount of training data.

-The working frequency range of the prototype mentioned in the paper is 4.5 GHz to 4.7 GHz, and there is a lack of analysis regarding its response under different incident angles. Could the author provide the simulation analysis for various incident angles?

-During the data collection phase, it is inevitable to encounter situations involving different angles, which often require manual intervention. Does the author have any automated data collection equipment in place?

-The author claims that the precision of the neural network's output has reached the level of noise. However, this may not necessarily be a positive outcome and could potentially be a result of overfitting the model. The data collected in a microwave chamber often differs from the noise present in real-world application environments, leading to increased inference errors. It would be helpful if the author could discuss this situation in their work.

-The author mentions, "In practice, we find each goal needs to be repeated for three to four times in order for the network to fully memorize it," which raises some confusion. Is this referring to the need for data repetition or a characteristic of the RNN? Additionally, when faced with such a significant data requirement, what advantages does the approach used offer compared to table lookup methods?

-The author mentions, "By using a large training set, the physical limit of the surface capability is approached." Could the author clarify what they specifically mean by the physical limit in this context and provide a clear inference?

Authors' Responses to Comments

Dear reviewers,

We would like to thank you for the incisive comments and constructive suggestions which greatly benefits the revised paper. Please kindly check attached the revised manuscript along with the following responses regarding your comments.

Reviewer #1

In this manuscript, the authors demonstrated an interesting way to design adaptive metasurfaces operating in microwave frequencies with a real-data-driven tandem artificial neural network. A 24-column 1D reconfigurable metasurface is fabricated for real-data acquisition. A generative-type neural network is firstly trained using some of the collected experimental data (80%), and then the performance of the trained neural network is tested using the rest of the data. The number of design targets can be as much as 5, and overall the performance of the device is very similar to the network design, confirming the great validity of the proposed method.

As compared with the previous result, this manuscript focuses on a regime (e.g, non-flat surface with a scatter near the surface) where the rational design method is hard to guarantee a good performance. From this point of view, this idea contains enough novelty and should be of great interest to researchers working in the field of optical communications and photonics. However, I would like to ask the authors to address the following questions before making the final decision:

1) I wonder if the authors can provide a numerical study to reveal how “bad” the conventional metasurface design method is in the situations reported in the text (i.e., non-flat surface, or perturbed scatter existed). If we just assume that there is no crosstalk between meta-atoms, we can map the target function with the meta-atom configuration with a simple Fourier transform. Of course, such a configuration may not work very well, but I hope to see how “bad” it is if we use the neural-network-predicted design as the bench marker.

This is a very good point. **We have added a section in Supplementary Note 3 on this.** We calculate the reflection pattern of a surface by treating it as an array and compare the result with simulation. This estimation method works well for flat surface as expected, but completely fail to match the result of a curved surface, missing a bunch of features in the reflection pattern. We believe this is an evident demonstration to show simple analytical model is indeed bad at calculating a complicated problem like this.

2) The authors focus on free-form designs in the manuscript, but I would say the well-defined target function could be more important in phased-array technologies. I am

sorry to say that, but to me, most of the designed far-field patterns seem like essentially Lambertian scattering with “random” noise, and I can hardly imagine and believe such a specific far-field pattern is the optimized solution for certain applications. I strongly recommend the authors use single-beam, dual-beam, and tri-beam steering as the target function to illustrate the power of the reported design principle. These are very basic, important, and easy-to-evaluate applications for phased-array technologies.

The original thought is that if the model works for random targets, it can perform beam-forming and null-forming which to some extent is a subset of it. But we do agree that the discussion on application was not very comprehensive in our first draft. **Thus, we have added more examples in beams/nulls steering results in Discussion and Extended Data Fig. 7.** We would like to keep the results for random targets since it showcases the free-form design capability of the proposed sequential network architecture, which can be important for other type of tasks or other engineering problems.

3) The authors demonstrated a 24-column 1D adaptive metasurface in the manuscript, which is good enough for a proof of concept. However, if we want to use this concept for practical, real applications, the dimension of the vector will become much larger (>1000). I guess that the computational cost is not growing linearly with the number of dimensions, and I wonder if the author could do some simple calculations and comment on the scalability of the proposed method.

We estimate that for the prediction network, since CNN is being used, it should have a scalability of $O(\log(n))$, and for the designer, a linear $O(n)$ relationship since it uses fully connected layer. However, providing evidence for this claim will need very comprehensive analyses and very carefully designed experiments, which is out of the scope of this work, but should make very good future study. **We have added this discussion in Supplementary Note 5.**

We would also like to point out that for controlling pattern in a single plane, 24 channel already works decently well in generating a few beams/nulls like shown in the revised draft, since it covers nearly 6 wavelengths.

4) The authors claimed the advantage a few times that they use measurement results instead of simulation results for data acquisition. My understanding is that people use simulation only when it is more convenient than doing real experiments. For example, much EM research is focused on passive device design, and it is impossible to fabricate millions of samples and test their performance one by one. I believe people make different choices in different situations, and the logic of some statements about this point in the manuscript could be improved. Perhaps the best way is to simply say “As we work with a reconfigurable design, actual experimental data is more convenient to be used for data acquisition, avoiding any possible errors that may arise from calculation, simulation or manufacturing tolerances.”

We agree that using real-data should not be considered as an advantage over other reported designs, but rather a practice happens to fit reconfigurable models. **We have deleted and rephrased relevant part in Introduction.**

Reviewer #2

This paper presented a deep-learning-enabled reconfigurable conformal metasurface. A flexible reconfigurable metasurface is presented and tested. The recurrent neural network is applied to achieve a neural-network predictor with customize input-size. However, considering the high quality of Nature Communications, I do not find enough innovation in this paper.

The authors declared that the reconfigurable flexible metasurfaces for curved surfaces are very challenging. However, some similar works have been achieved many years ago [1-4]. The metasurface designed in this paper does not show any advantages in terms of amplitude loss and bandwidth. More recently, the flexible RF circuits with more compact size and functions have been proposed [5,6]. Thus, I do not find innovation of the metasurface designs in this paper.

It might not be clear enough in our first draft that the innovation here is to solve the RF high Q (loss) issue for tunable flexible metasurface in microwave frequency, rather than implement the first flexible surface. **We have modified the Introduction and Section 1 to make this more clear.**

In fact, the very limited operational scenario/compromises of the designs in those reference just showcases this drawback:

[1] [2] [4] verify the point that the loss of thin metasurface is high so that the application is limited to less loss-sensitive tasks like reducing reflections.

[3] try to avoid loss by avoiding electronic parts, which is far not comparable to electrical tuning devices in terms of accuracy and response speed.

[5][6] doesn't seem to be relevant since they are not dealing with RF signals.

In short, none of past studies actually deal with this critical loss issue. **We have cited some of them in Introduction to demonstrate this point.**

It is also worth noting that the Q-volume limitation is an essential limitation (Chu's limit) rather than only for our design, so the general methodology of a double-stack solution that compromises neither physical volume nor flexibility can only provide benefit to any given designs. **We have added a section in Supplementary Note 3 to support this point.**

Another advantage declared by the authors is using the measured data for neural network training (called as 'real-data-driven'). As the numeric simulations require matrix calculations, the direct measurement is surely faster than the simulations. However, the

idea of using real-data is not new and has already been used for various reconfigurable metasurface imaging and radar detections [7, 8]. The authors did not demonstrate more breakthrough methods and theories in this paper.

We agree that using real-data should not be considered as an advantage over other reported designs, but rather a practice happens to fit reconfigurable models, as also pointed out by Reviewer #1. **We have deleted and rephrased relevant part in Introduction.**

Another main innovation the reviewer didn't mention here, which is also the main breakthrough of this work, is the design of a universal inverse-design algorithm with free-form design capability. As far as we are concerned, this is the main challenge (along with the loss issue mentioned above) that keeps past conformal metasurfaces from being used for complicated tasks with random shapes. (This can be seen from the reference provided by the reviewer: the most common setup is simple cylindrical surface used as absorber). To our best knowledge, there were no approach that ever enables any given irregular-shaped surface to perform such versatile tasks like beam steering, null forming, even cloaking object in front.

Combining the innovations mentioned above, and considering the very essential and important function it achieves (along with the potential that the proposed NN architecture could be used in any other engineering/science fields), we believe this work is significant for the community and is likely to draw very board attention.

Two advices:

1. The symbolic representation in this paper is confusing. In page 5, "a sequence with a length of l_t , repeating n_t angle-directivity pairs (θ_i, D_i) ", the definition of subscript 't' and 'i' should be explained. In Fig.2b, the input of the designer is $(\theta_i, D_{\theta i})$, however, in Extended Data Fig. 1, the input of the designer is (θ_i, D_i) , which is not uniform. Meanwhile, Fig. 2b displays an autoencoder structure made of designer and predictor, which means the output of the predictor (D_i) should correspond to the input of designer $(\theta_i, D_{\theta i})$. The authors should explain the correspondence between D_i and $D_{\theta i}$, better shown in the figure.

Thank you for pointing this out! The notation in the figures and text are indeed not coherent in our first draft. **We have fixed the notation in Fig. 2, Extended Data Fig.1 and text in Section 3.**

2. In page 8, the authors mentioned that "Therefore, this trained network can be deployed on modest micro-controllers with very limited computing power." However, the efficiency of the designer network hasn't really been verified in a micro-controllers nor embedded in a system-level application. As reference works, a deep-learning-enabled self-adaptive metasurface clock deployed on micro-controllers is displayed in [R1], and an intelligent metasurface system for automatic tracking of moving targets and

wireless communications deployed on FPGA is displayed in [R2]. The authors should add [R1] and [R2] in the References and discuss the advantages of this paper compared with them.

The speed of the network prediction is verified on a modest micro-controller Raspberry Pi in Table 1, the results show a response time on the order of ms. Compared with this, other process in the system like incident angle and frequency sensing are relatively trivial for our case, and should be built according to different specific operating environment and setups.

[R1] was in the original draft and we have added [R2] in the revision.

[R1] QIAN C, ZHENG B, SHEN Y C, et al. Deep-learning-enabled self-adaptive microwave cloak without human intervention. *Nature Photonics*, 2020, 14(6): 383-+.

[R2] LI W, MA Q, LIU C, et al. Intelligent metasurface system for automatic tracking of moving targets and wireless communications based on computer vision. *Nature communications*, 2023, 14(1): 989-.

References:

- [1] Park, E., Li, W., Jung, H., Lee, M., Park, J.-H., Shamim, A., Lim, S., *Adv. Mater. Technol.* 2023, 8, 2201451.
- [2] X. -Y. Luo et al., "Active Cylindrical Metasurface With Spatial Reconfigurability for Tunable Backward Scattering Reduction," in *IEEE Transactions on Antennas and Propagation*, vol. 69, no. 6, pp. 3332-3340, June 2021, doi: 10.1109/TAP.2020.3037728.
- [3] X. D. Liang and A. J. Crosby, "Uniaxial stretching mechanics of cellular flexible metamaterials," *Extreme Mechanics Letters*, vol. 35, Feb 2020, Art no. 100637.
- [4] S. Liu, H. Xu, H. Zhang, and T. Cui, "Tunable ultrathin mantle cloak via varactor-diode-loaded metasurface," *Opt. Express* 22, 13403-13417 (2014).
- [5] G. Li et al., "Three-dimensional flexible electronics using solidified liquid metal with regulated plasticity," *Nature Electronics*, vol. 6, no. 2, pp. 154-163, 2023/02/01 2023.
- [6] Y. Bai et al., "A dynamically reprogrammable surface with self-evolving shape morphing," *Nature*, vol. 609, no. 7928, pp. 701-708, 2022/09/01 2022
- [7] L. Li et al., "Machine-learning reprogrammable metasurface imager," *Nature Communications*, vol. 10, no. 1, p. 1082, Mar 6 2019, Art no. 1082, doi: 10.1038/s41467-019-09103-2.
- [8] S. Wei et al., "3DRIED: A High-Resolution 3-D Millimeter-Wave Radar Dataset Dedicated to Imaging and Evaluation," *Remote Sensing*, vol. 13, no. 17, 2021, doi: 10.3390/rs13173366.

Reviewer #3

Manipulating the electromagnetic (EM) scattering behavior dynamically is an ambitious objective in the fields of EM stealth and communication. However, conventional analysis

and optimization techniques fall short in accomplishing this task. In this paper, a reconfigurable conformal metasurface prototype is introduced, capable of responding to various design targets in reflection patterns with minimal on-site computing power and time. The metasurface utilizes a sequential tandem neural network that is pre-trained with real experimental data, eliminating errors from calculations, simulations, or manufacturing tolerances. I conclude the innovations of this work as two parts, the realization of conformal metasurfaces and on-site machine learning. I need to make further consideration after the authors solve the following problems.

-The tunable conformal metasurfaces are demonstrated in one dimension. How to realize two-dimensional case, which is more useful.

Good point. The same stack-up methodology can be used for other unit designs, like unit design in [1], where all lumped component is inside a unit. For the algorithm, 1-D CNN can be replaced by 2-D CNN with very similar manner. **We have added this in the Discussion.**

[1] QIAN C, ZHENG B, SHEN Y C, et al. Deep-learning-enabled self-adaptive microwave cloak without human intervention. *Nature Photonics*, 2020, 14(6): 383-+.

-Tandem neural network has been studied before to imitate the non-uniqueness issue in inverse design. In this work, what's the new points in sequential tandem neural network.

For tandem networks, a design target with the exact form as the predictor's output is required as the input of the network, for example, a vector that contains the whole pattern with, say, 5 or 10 degrees of resolution are needed. This largely limits its practicality for cases where free-form design goals are more favorable, for example, one may want to specify several beam/null directions without having to constructing the entire pattern. **We have expended the discussion in Section 2 to make it more clear.**

-For each scene, the authors need to train the neural network case-by-case. Although the authors clarify that they use experimental data and on-site training, the case-by-case neural network is inefficient. In other words, if it is possible to build up the neural network connection between different scenarios to speed up network convergence and save the amount of training data.

This work can be used for most application consisting of non-moving part, like buildings exteriors, airplane skins, etc. For moving part, if the shape can be parameterized (like airplane wings that has only several degrees of freedom), that information can potentially be input into the network. **We listed this possibility as a future work in the Discussion.**

-The working frequency range of the prototype mentioned in the paper is 4.5 GHz to 4.7 GHz, and there is a lack of analysis regarding its response under different incident angles. Could the author provide the simulation analysis for various incident angles?

Some previous studies pointed out that for electric small ($<1/3$ wavelength) element under small incidence (<45 degree), the specular reflection effect from single unit is insignificant (single unit has low directivity compared to array factor). We are using a small element ($\sim 1/4$ wavelength) so the pattern should mostly be determined by phase difference between elements. Nevertheless, for data-driven models, none of these analyses need to be taken, which is a main advantage of data-driven model.

-During the data collection phase, it is inevitable to encounter situations involving different angles, which often require manual intervention. Does the author have any automated data collection equipment in place?

The whole measurement process is automated without human intervention, achieved by a single NI controller. **We have made this point more clear in the caption for Fig.3.**

-The author claims that the precision of the neural network's output has reached the level of noise. However, this may not necessarily be a positive outcome and could potentially be a result of overfitting the model. The data collected in a microwave chamber often differs from the noise present in real-world application environments, leading to increased inference errors. It would be helpful if the author could discuss this situation in their work.

We believe this is not overfitting since all visual examples and statistical analysis are on validation set (unless specified), which the network never seen in training process. We do agree that the chamber may still not be ideal for data collection (but this is the one most close to ideal environment we have access to). For more accurate readings, or complicated environments, the data can be taken with proper equipment available under environment that most resembles its operating condition. **We have added this discussion in the caption of Fig.3.**

-The author mentions, "In practice, we find each goal needs to be repeated for three to four times in order for the network to fully memorize it," which raises some confusion. Is this referring to the need for data repetition or a characteristic of the RNN?

A more rigorous and less confusing statement might be, "in practice, we found the sequence length I_t should be three or four times of target number N_t to yield a converged results for designer, thus for up to 5 targets, we choose $I_t = 20$ ". **We have rephrased this in Section 2.**

Additionally, when faced with such a significant data requirement, what advantages does the approach used offer compared to table lookup methods?

Though 260,000 seems to be a large number it is absolutely not comparable to dataset needed for building a table lookup table. A simple calculation: a most coarse grid search with a resolution of 0.2V (very reasonable from the reflection curve) from 10V to 18V needs $40^{24} = 2.8e38$. **We have added this discussion in Section 3.**

-The author mentions, "By using a large training set, the physical limit of the surface capability is approached." Could the author clarify what they specifically mean by the physical limit in this context and provide a clear inference?

The physical limit was meant to refer to the max achievable directivity, beamwidth limit, etc. From Extended Data Fig.7 (newly added in revised draft), it seems to be the case for single beam forming. But you are right that we did not rigorously prove the network can reach all physical limits, and it makes very good future topic to examine whether and how these physical limits can be approached. **To be fair, we have deleted this statement in the paper.**

REVIEWER COMMENTS

Reviewer #1 (Remarks to the Author):

I appreciate the authors for their great efforts in addressing my questions and concerns about this manuscript. The additional simulations and analysis shown in extended Fig. 7 and Supplementary Notes 3 and 5 have excellently answered the questions to which they know the exact answer. I also appreciate their rigorous scientific attitude in deleting/revising the unclear contents in the revised manuscript.

Overall, I am satisfied with the revised manuscript, and I can now recommend the publication of the paper.

Reviewer #2 (Remarks to the Author):

I read the revised manuscript and added information carefully, and examined the replies to my concerns and other reviewers' comments, and found that the manuscript was well revised. I suggest publication of the new version.

Reviewer #3 (Remarks to the Author):

In assessing this manuscript, we have noted that the authors have not provided sufficient data and theoretical support to explain and substantiate their responses in their replies. Their reliance on textual descriptions without substantial backing is evident in response to comment 1, 3, 4, 6 and 8. Notably, in question 4, the authors reference some prior research results but fail to supply relevant citations, which diminishes the credibility of their responses. Additionally, we believe that the unit responses of the metasurface may vary at different incident angles, potentially impacting its performance in modulation. Therefore, we request the authors to provide corresponding simulation results to support their claims. However, the authors have not adequately addressed this point or provided a reasonable response.

In conclusion, given the aforementioned issues, we cannot recommend further publication of this manuscript unless the authors are able to readdress all the questions and furnish substantial data and theoretical support to rectify these deficiencies. We hope the authors will earnestly consider these review comments and make further improvements to their research in future work.

Authors' Responses to Comments

Dear reviewers,

We would like to thank you for your time and effort making our draft better. Please kindly check attached the revised manuscript along with the following responses regarding your comments.

Reviewer #1

I appreciate the authors for their great efforts in addressing my questions and concerns about this manuscript. The additional simulations and analysis shown in extended Fig. 7 and Supplementary Notes 3 and 5 have excellently answered the questions to which they know the exact answer. I also appreciate their rigorous scientific attitude in deleting/revising the unclear contents in the revised manuscript.

Overall, I am satisfied with the revised manuscript, and I can now recommend the publication of the paper.

Thank you for your precious advice that help improve the credibility of the draft significantly!

Reviewer #2

I read the revised manuscript and added information carefully, and examined the replies to my concerns and other reviewers' comments, and found that the manuscript was well revised. I suggest publication of the new version.

Thank you for your advice to our draft and your appreciation to the merit of our work!

Reviewer #3

In assessing this manuscript, we have noted that the authors have not provided sufficient data and theoretical support to explain and substantiate their responses in their replies. Their reliance on textual descriptions without substantial backing is evident in response to comment 1, 3, 4, 6 and 8. Notably, in question 4, the authors reference some prior research results but fail to supply relevant citations, which diminishes the credibility of their responses. Additionally, we believe that the unit responses of the metasurface may vary at different incident angles, potentially impacting its performance in modulation. Therefore, we request the authors to provide corresponding simulation results to support their claims. However, the authors have not adequately addressed this point or provided a reasonable response.

In conclusion, given the aforementioned issues, we cannot recommend further

publication of this manuscript unless the authors are able to readdress all the questions and furnish substantial data and theoretical support to rectify these deficiencies. We hope the authors will earnestly consider these review comments and make further improvements to their research in future work.

We feel sorry that our first revision did not fully resolve your concerns and fulfill your expectation. Please kindly find the following our attempt to readdress these issues, we hope these revision shows our resolution in making the draft very credible:

Original question 4: The working frequency range of the prototype mentioned in the paper is 4.5 GHz to 4.7 GHz, and there is a lack of analysis regarding its response under different incident angles. Could the author provide the simulation analysis for various incident angles?

Sorry that we might misunderstand the original question. We were under the assumption that the question was about the modulation effect of the pattern of a single unit.

Regarding to the dependency of unit's response to incident angle, **we have added following results in Supplementary Note 3:**

For E-plane oblique incidence, since it relates to the validity of the data gathering setup using specular reflection, **we simulate the unit under the 25-degree incidence (as in the setup)**, we do not observe noticeable difference compared to normal incidence.

For H-plane oblique incidence, **we simulate the unit under normal,30,45 degree and observe slight difference when incident angle is large.** However, we want to note again that those frequency response results are only used to verify the surface's capability of generating different reflection phases, but are NOT used for the inverse design algorithm. In other words, the 'modulation effect' is already baked in the neural network which is trained with real measurement data of the whole surface under different incident angles.

Thus, these new findings do not undermine the original performance/claims. On the contrary, the incidence dependency in H-plane provides another factor that complicates the analytical approach, since in curved surface each unit has different local incident angle. **We have also added this point in Chapter 1.**

Original question 1: The tunable conformal metasurfaces are demonstrated in one dimension. How to realize two-dimensional case, which is more useful.

To further demonstrate the possibility of accommodating the surface to a 2-D version, **we have added a new Chapter in the Supplementary Notes.** We take the design in Ref [30], create boarder without rigid substrate and tune the dimensions for it to work around the frequency as in our study. We believe this is clear evidence that the accommodation is very realizable.

Original question 6: The author claims that the precision of the neural network's output has reached the level of noise. However, this may not necessarily be a positive outcome and could potentially be a result of overfitting the model. The data collected in a microwave chamber often differs from the noise present in real-world application environments, leading to increased inference errors. It would be helpful if the author could discuss this situation in their work.

In our original answer, we gave evidence that the network itself is not overfitted (good performance in validation set), and we assume the reviewer's remaining concern might be: a noise-level error in the chamber does not guarantee a good error in free-space. Regarding this, **we have added some calculation in Supplementary Note 4** to show that the SNR in chamber is estimated to be at least 22 dB even before calibration, and expected to be above 30 dB after calibration. Along with the fact that using chamber is the most common practice to emulate free-space environment in metasurface studies [6,13,14-16, 26, 27, 30, 31], we think it is fair to assume the results in chamber to be a good enough representation of free-space and to claim the excellent accuracy of the network. To be rigorous, **we have also rephrased some of the statements and added some description in Chapter 3.**

On other questions, please find following explanations why we believe textual revision is enough:

Original question 3 - For each scene, the authors need to train the neural network case-by-case. Although the authors clarify that they use experimental data and on-site training, the case-by-case neural network is inefficient. In other words, if it is possible to build up the neural network connection between different scenarios to speed up network convergence and save the amount of training data.

Original answer - This work can be used for most application consisting of non-moving part, like buildings exteriors, airplane skins, etc. For moving part, if the shape can be parameterized (like airplane wings that have only several degrees of freedom), that information can potentially be input into the network. We listed this possibility as a future work in the Discussion.

Though being a very inspiring topic, implementing a whole different working scenario with moving part is out of the scope of this work. We believe the real-time inverse-design for a stationary surface (which already suitable for many working conditions) is already a huge step forward from previous conformal surface studies, and is a significant first step towards more advanced forms of conformal surfaces, which also include ones covering moving structures. Thus, we think it is reasonable to leave it as a future work without going deep into implementing it.

Original question 8 - The author mentions, "By using a large training set, the physical limit of the surface capability is approached." Could the author clarify what they specifically mean by the physical limit in this context and provide a clear inference?

Original answer - The physical limit was meant to refer to the max achievable directivity, beamwidth limit, etc. From Extended Data Fig.7 (newly added in revised draft), it seems to be the case for single beam forming. But you are right that we did not rigorously prove the network can reach all physical limits, and it makes very good future topic to examine whether and how these physical limits can be approached. To be fair, we have deleted this statement in the paper.

Due to difficulty in finding well-defined metrics to evaluate to what extent this limitation is approached, we already deleted this claim in our first draft. Thus, we do not see possibility to provide any evidence to support any claim.

REVIEWERS' COMMENTS

Reviewer #3 (Remarks to the Author):

I have carefully reviewed the latest manuscript and am satisfied that the relevant issues I raised have been addressed. I recommend the publication of the latest version of the manuscript.

Authors' Responses to Comments

Dear reviewers,

We would like to thank you for your time and effort making our draft better. Please kindly check attached the revised manuscript along with the following responses regarding your comments.

Reviewer #3

I have carefully reviewed the latest manuscript and am satisfied that the relevant issues I raised have been addressed. I recommend the publication of the latest version of the manuscript.

We are glad our revision resolves your concerns. We thank you for your comments that greatly improve the credibility of this work.